# The Influence of the Use of Polymer Lining within the Roller Press Gravity Feeder on Briquette Quality

**DOI:** 10.3390/polym12112489

**Published:** 2020-10-27

**Authors:** Michał Bembenek

**Affiliations:** AGH University of Science and Technology, Faculty of Mechanical Engineering and Robotics, Department of Manufacturing Systems, 30-059 Kraków, Poland; bembenek@agh.edu.pl; Tel.: +48-12-617-31-20

**Keywords:** roller press, material flow, polymer lining, polymer layers, hoper improving, briquetting

## Abstract

When considering the operation of roller presses for the consolidation of fine-grained materials, the main problems are disturbances in the proper flow of the material and its bridging in gravity feeders. This is especially true for small and medium capacity presses, where the hoppers for dosing the material are narrow. This article presents innovative laboratory tests of the impact of using a polymer plate lining in the gravity feeder of a roller press. Polymer materials Polyacetal C (POM C) and Ultra-High-Molecular-Weight Polyethylene (UHMW-PE) were used for the tests. The influence of the use of plates on the material flow and quality of briquettes was investigated in comparison with the case where such plates were not used. The research showed an improvement in the flow of fine-grained materials in the feeder and an increase of the briquette strength indexes, as compared to those cases when polymer linings were not used in the feeder.

## 1. Introduction

Loose materials are often given a form of compact pieces because this is the form in which they are usable in many engineering processes [1], e.g., biomass processing [2] or fertiliser production [3]. The consolidation process offers a great number of advantages such as easier transport, increased economic efficiency of processes [4] or improvement of performance properties by giving proper grain size to the product [5]. The agglomeration process makes it possible to process raw materials, manufacture new products [6] and utilise waste [7]. Roller presses are the types of machines which are most frequently used in the pressurised agglomeration process [8], especially for the consolidation of other materials than plant materials [9]. These machines distinguish themselves for their constant operability with relatively low demand for energy and a longer life span of the moulding components as compared to other briquetting machines, e.g., screw or stamping ones. Roller briquetters are machines that are designed and made for strictly defined purposes [10]. This is due to the properties of the input materials and the purpose of products of consolidation [11]. A standard roller press design usually contains the following subassemblies: drive system, working rollers cage, roller support system and compaction unit. The drive system consists of a motor or motors and torque transmission elements, that is, various kinds of clutches and gears coupled with the compaction unit [12]. The compaction unit consists of working rollers and a material feeding unit. It is situated in a working rollers cage and is protected against overloads by the so-called roller support system. Working rollers cages are built of frames with various design solutions. The briquette forming process takes place between two rollers which rotate in mutually opposite directions and on the surfaces of which there are moulding cavities distributed in a proper way [13]. These are semi-moulds between which material supplied from a hopper is moulded into briquettes. To obtain a correct quality product, it is necessary to ensure a relevant material compaction degree after consolidation. The basic solution ensuring proper flow and dosage of material into the compaction zone is the gravity feeder. The material moves down the feeder under the gravity force towards moulding rollers. The characteristic features of this solution include simplicity of design and lack of additional external power supply, which also makes it the cheapest solution. The gravity feeder may only be used for materials which do not show a tendency to hang up and form vaults. If the feed tends to hang up in the feeding system or must be pre-compacted, then material flow-enforcement feeders are used, e.g., a screw feeder [11]. It also enables a significant increase of the unit pressure exerted in the briquetting process, the maximum value of which may even be twice higher as compared to the pressure generated when using gravity feeder [14]. It should be mentioned, however, that the use of flow-enforcement feeders requires the press design to be expanded and complicated. This also increases the press purchase and maintenance costs. This results from the additional consumption of electrical energy to power drive motors of flow-enforcement feeders [15]. Therefore, in case of materials, the flow of which is hindered in gravity feeders, additional equipment improving flow fluency can be used [16]. Similar issues and solutions related to the improvement of the hindered flow of fine-grained material are related to many sectors of the industry [17], e.g., food [18], pharmaceutical [19], comminution [20], etc. They are being developed especially for hoppers [21], bunkers and siloes [22] for loose materials commonly supported with a computer method [23] like DEM [24] or image analysis [25].

The basic action taken to improve the flow and unloading is to properly design the geometry of the tank [26], e.g., wedge [27], conical [28], angled [29], hour-glass, clepsydra [30], pyramidal [31] tilted [32], curved shaped [33] and its outlet [34]. The hopper should be adequately matched, taking into account the type of material, the nature of its flow and its properties [35] like grain size [36] and shape [37], vertical pressures and compressive friction force [38] or flow properties [39]. As tests have shown, a proper shape of the tank inlet or using a proper insert [40] or rotating orifice [41] also has an effect on correct emptying [42]. Numerous mechanical solutions [43], e.g., pneumatic or mechanical vibrators, are also employed [44]. Vibrators can be divided into designs with a linear and circular nature of vibrations. The operating agent of pneumatic vibrators is compressed air supplied from a separate compression system. In case a solution of this type is installed, the gravity feeder structure should be provided with damping components installed to prevent an excessive propagation of vibrations to other components of the press. Another solution employed is pneumatic hammers [45]. While operating, hammers release mechanical energy which is transferred through the feeder walls at equal time intervals. A series of impacts, the power of which can be controlled by changing the air pressure in the system, causes the material deposits accumulating on walls to be removed and prevents material bridges from being formed inside the feeder. The hammers are installed by means of special brackets. If the performance specifications of the hammers are wrongly set, this may cause the flow to deteriorate because material bridges are tamped instead of being broken apart. A third solution being employed is aeration nozzles [46]. They are installed on tank walls. Air pressure is applied directly on the material by means of silicone edges adjoining the feeder inner walls [47]. Aeration of material prevents it from hanging up inside and facilitates the flow in the compaction zone [48]. In order for this mechanism to work effectively, nozzles must properly distributed in a proper number to ensure a proper material flow. Due to the decrease of bulk density of the material being consolidated, this solution is not recommended for roller press feeders.

Another way to improve the flow of material in the roller press compaction system is to line up the hopper internal walls with boards made of material characterised by low friction coefficient, low adhesion, high resistance to mechanical wear, high impact strength and resistance to chemicals. The boards must have proper shape, dimensions and thickness so as to fully cover the inner walls of the hopper, thereby having no effect on the movement of materials and, in the case of roller presses, on the rotational movement of working rollers. As is the case for siloes, using the boards for narrow gravity feeders should result in a better flow of material without bridges being formed. In case of wide feeders, i.e., those more than 500 mm wide, there may be one more advantage coming along with the use of slide boards. The unit pressure exerted in the moulding cavities is not identical along the entire forming length of the roller. Peak unit pressure is reached in the middle of the roller because the flow of the material being consolidated is most intense in this place. The slowest flow of material occurs at the walls of the feeder side sealings. This results from the friction force acting between the material and the feeder walls, which causes the flow to brake in these zones. This is the reason why the wear of the rings increases away from the feeder walls towards their centre [11]. The outermost cavities show the slowest wear rate, which results in the rings showing their highest degree of wear in their central part. This causes the diameter of the rings in their central part to decrease. Therefore, it can be concluded during the long-lasting operation of roller presses that the improvement of the flow of material being compacted on side walls of large width feeders may lead to a more uniform wear of the working rollers.

This article presents innovative laboratory tests of the effect of the polymer board lining being used in the gravity hopper of the roller press. Materials recommended for silo linings, i.e., POM C and UHMW-PE, were used for the tests. The effect of the use of boards on the flow of material and the quality of briquettes was tested as compared to a case where such boards are not used was tested. The presented research is novel and innovative and there no data in the literature about this topic. The results can be implemented to a new construction of roller presses.

## 2. Materials and Methods

Research of the use of polymer lining within the roller press gravity feeder on briquette quality was done using a roller press (Figure 1) with a 450 mm roll pitch diameter with an installed compaction unit. It produces the saddle-shaped briquettes with a size of 31 mm × 30 mm × 13 mm and a rated capacity of 6.5 cm^3^ (Figure 2). The outline view of the moulding surface used to consolidate the material is presented in Figure 3. The press was equipped with a cycloidal gear motor with a power of 22 kW and a frequency converter that enabled infinitely variable control of the revolutionary speed of the rolls. All materials were consolidated using a gravity feeder with a roller revolutionary speed of 4.25 rpm, which corresponded to the peripheral speed of the rolls equal to 0.1 m/s with an inter-roll gap of 1 mm.

Two polymer composites were selected for laboratory tests: POM C (Polyacetal C) and UHMW-PE (Ultra-High-Molecular-Weight Polyethylene). Both materials are characterised by resistance to abrasive wear and high impact strength. They are resistant to weather conditions and neutral to materials which are consolidated in roller presses. Polyacetal POM C belongs to a group of thermoplastic materials with a semicrystalline chemical structure. Thanks to its properties, it is widely used in many sectors of industry. It is used to manufacture high-quality components of machines and structural elements such as toothed gears, bearings, levers and cams. UHMW-PE belongs to flexible thermoplastics. It is a composite which is used in the industry in the form of boards making it easier for the material to flow in large siloes and tanks. UHMW-PE is often used in the food and chemical industry to manufacture, among others, tanks, siloes, pipes and toothed gears. Table 1 presents basic parameters of both composites. Board surface roughness tests were also carried out. The TOPO L50 modular system for the measurement and analysis of surface topography was used to measure the surface roughness in directions which are perpendicular to the surface structure direction based on the R profile type and the length of 5 elementary sections.

Four materials which represented three groups—materials of inorganic origin, materials of organic origin/fuels and heavy industry waste—were used for tests:
materials of inorganic origin:hydrated lime,artificial fertiliser for lawns,material of organic origin/fuels: charcoal,heavy industry waste:electric arc furnace dust (EAFD) mixtures.

Materials of various origin and various chemical composition and structure, so as to make it possible to carry out experiments for diverse material properties, were selected. Except for the artificial fertiliser, the materials selected for the test tend to hang up or irregularly flow in hoppers. Before the consolidation process, six mixtures of the materials referred to above were prepared. They were thoroughly mixed and brought to a proper moisture enabling them to be consolidated in a roller press. Binders were added. Some materials were heated up to 70 °C. Moisture was determined by the weight method at 105 °C until a constant weight was obtained. The Vibra AJH 420 CE (Tokyo, Japan) scale was used.


**Mixture 1.**


Its composition was 83.3% calcium hydroxide manufactured by Lhoist (EN 459-1 CL 90-S) (Limelette, Belgium) and 16.7% water. The mixture was mixture in Z-mixer for about 30 min. The moisture content of the mixture was 16.9%.


**Mixture 2.**


It was the granules from fertiliser “nawóz do trawników” produced by Agrocel Company (Babimost, Poland) with a grain size down to 3 mm. The fertiliser contents: N—19%, P_2_O_5_—5%, K_2_O—9%, MgO—2%, and Fe—1% [51]. The moisture content was as low as 0.1%.


**Mixture 3.**


It was the charcoal fines mixed with a 4% of starch heat up to 70 °C. The moisture of the mixture was 26.9%. The mixture temperature during briquetting 70 °C—semi-hot briquetting.


**Mixture 4.**


It was the charcoal fines mixed with a 6% of starch heat up to 70 °C. The moisture of the mixture was 27.1%. The mixture temperature during briquetting 70 °C—semi-hot briquetting.


**Mixture 5.**


It was the charcoal fines mixed with a 6% of starch. The moisture content of the mixture was 27.1%. The mixture temperature during briquetting 21 °C.


**Mixture 6.**


The mixture contained 47.7% of EAFD, 36.7% of scale, 7.3% fine coke breeze, 5.5% 80°Bx molasses and 2.8% calcium hydroxide. The last two ingredients acted as a binder [7]. The mixture was mixed in a Z-mixer. Its moisture content was 4.6%.

The polymer lining of the gravity feeder has been obtained by designing and making specifically shaped inserts (Figure 4) of polymer materials selected for tests. The inserts were made of 6 mm thick boards available on the market. Depending on the series of tests, they were installed or not in the LPW 450 press gravity feeder (Figure 5). The prepared mixtures were divided into three samples. Each mixture was briquetted in a roller press equipped with a gravity feeder with POM C lining, UHMW-PE lining and without lining. A total number of 18 briquetting tests were done.

After the briquetting process, the drop strength and compressive strength of the selected briquettes were tested. The briquettes drop strength consisted of simultaneously dropping 10 random selected briquettes three times from the height of 2 m onto a steel plate. Then, they were sifted through a sieve with the mesh size equalling 2/3 of the average of the 2 largest dimensions of the briquette. The value of the drop strength is represented by the percent share of the sifting in the total weight of the briquette sample. The compressive strength is a value of the force destroying briquette which is determined in course of a uniaxial compression test, performed between two parallel flat surfaces. It was done by Zwick 1120.20 (Ulm, Germany) equipped with the 2 kN head. Ten briquettes were tested and 2 extreme samples were discarded; the mean value was calculated. Strength tests of the obtained briquettes were carried out one day and one month after the briquetting process. The briquettes were seasoned at an ambient temperature equal to 21 °C.

## 3. Results

The results of measurement of surface roughness of POM C and UHMW-PE are presented in Table 2.

During the experiments in which the polymer lining was installed on the gravity hopper walls, an improved flow of mixtures in the compaction zone was observed as compared to the tests where no lining was installed. This refers both to the lining made of boards made of POM C as well as UHMW-PE. Particularly positive flow improvement results were noticed during the briquetting of hydrated lime and charcoal in the cold form. Both materials tended to hang up on the feeder walls when the polymer lining was not used. The input material was found to move with difficulties and to tend to hang up also during the warm briquetting of charcoal with a 4 and 6% starch additive. These phenomena have been partly eliminated in case of tests with the installed lining. While briquetting artificial fertiliser and EAFD mixtures, no tendencies to hang up were observed when boards were used and when they were absent. Both input materials fluently moved into the compaction zone. Small differences were observed in the movement of materials in the compaction zone with boards made of the POM C and UHMW-PE material installed. The POM C material showed slightly better effects in terms of the flow fluidity but these differences were often visually unnoticeable in case of most input materials. The results of the strength tests of the selected briquettes obtained are shown in Table 3 and on Figure 6, Figure 7, Figure 8, Figure 9, Figure 10, Figure 11, Figure 12 and Figure 13. The compressive strength tests of the charcoal mixture could not be performed on the next day after briquetting due to the plastic nature of the briquettes.

## 4. Discussion

The performed tests have an innovative nature. The polymer materials selected for slide boards complied with our assumptions. The observations made during the tests and the analysis of results have shown that the effect of polymer linings on the quality of briquettes is positive. In most cases, they made it significantly easier for the input material to move in the compaction zone, which resulted in, among others, better compaction of the material and so in the improvement of performance values of the briquettes produced. Despite the fact that the polymer linings made of the POM C material were characterised by nearly three times higher surface roughness (R_a_ = 2.56 µm, R_z_ = 13.88 µm), they turned out to have a better effect on the quality of the briquettes produced. Table 4 presents a percent increase of the average compressive strength and the average drop strength of the briquettes produced with polymer linings installed on the hopper walls against the values obtained for briquettes with not linings.

Following the analysis of the results obtained, it can be concluded that improved strength parameters were obtained in all cases where the POM C polymer was lined up. The improvement of indicators ranged from 0.1 to 88.4% for the drop strength and from 10.7 to 99.2% for the compressive strength, which we consider to be very good results. For UHMW-PE, the drop strength change ranged from −1.8 to 34.9%, while the compressive strength change ranged from −5.9% to 60.3%, which can also be considered to be good results, with the negative results considered to be of minor importance due to their value being low. The best improvement of the briquette strength indicators was obtained for a calcium hydroxide mixture. This is a raw material which causes most problems during consolidation due to its properties, including low bulk density and a tendency to form bridges. The smallest impact of the use of polymer linings on the briquettes quality was observed for Mixture 3 (charcoal fines mixed with 4% of starch). However, it should be noted that the mixture prepared for briquetting was heated up to 70 °C. This likely caused the contained heated binder in the form of starch dissolved in water to adhere more closely to the liners of the feeder. Taking all test results obtained and the above-mentioned mechanical properties, including the yield point of the POM C polymer, into account, it will presumably be a better and more durable material. At present, the disadvantages of using this material as a lining for the feeders are not noticed.

## 5. Conclusions

The innovative solution involving the use of polymer linings in roller press hoppers is worth paying attention to and being further analysed. It allows to obtain briquettes with better strength properties; furthermore, it improves the movement of materials within the compaction zone while eliminating the need to use other types of solutions. The simple design, low manufacturing costs and easy installation of the boards without interfering with the readymade structures of roller presses makes this solution unique and applicable to a wide range of consolidated materials and press designs.

In future tests, it would be necessary to test slide boards for their durability when being operated for a longer period of time. The effect of the linings can also be verified for a broader range of materials being briquetted and for roller presses with wider charging inlets.

## Figures and Tables

**Figure 1 polymers-12-02489-f001:**
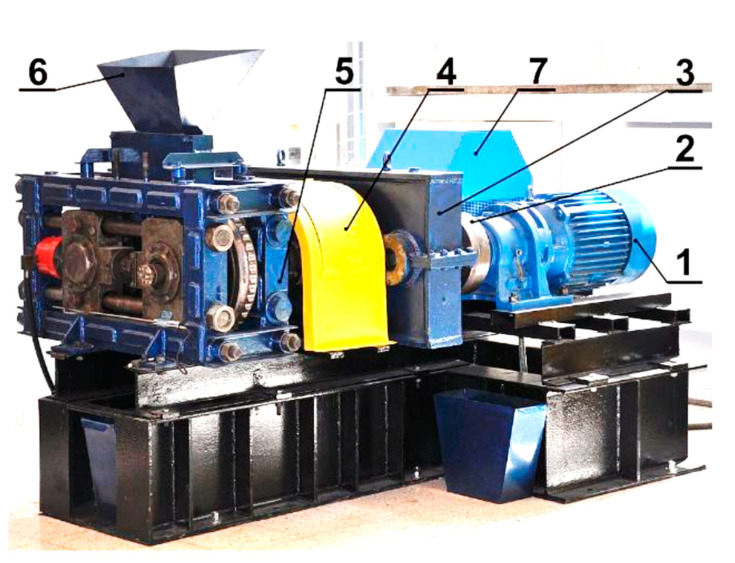
LPW 450 laboratory roller press: **1**—gear motor with a cycloidal transmission, **2**—flexible clutch, **3**—gearbox, **4**—enclosure of Oldham couplings and friction clutch, **5**—moulding rollers cage, **6**—gravity feeder, **7**—hydraulic system of sliding roller support.

**Figure 2 polymers-12-02489-f002:**
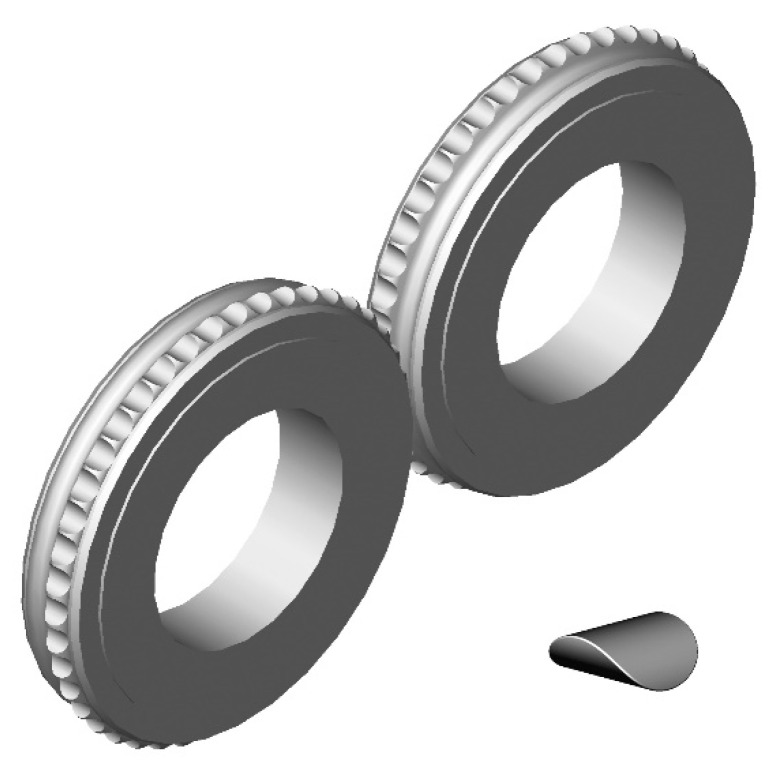
Moulding rings used in roller press compaction unit with a saddle-shaped briquette.

**Figure 3 polymers-12-02489-f003:**
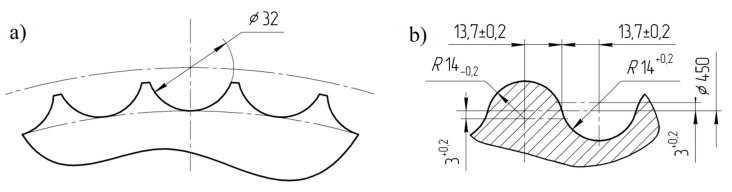
Geometry of moulding cavities on the working surface of rolls used for tests: (**a**) front view and (**b**) cross section through the groove.

**Figure 4 polymers-12-02489-f004:**
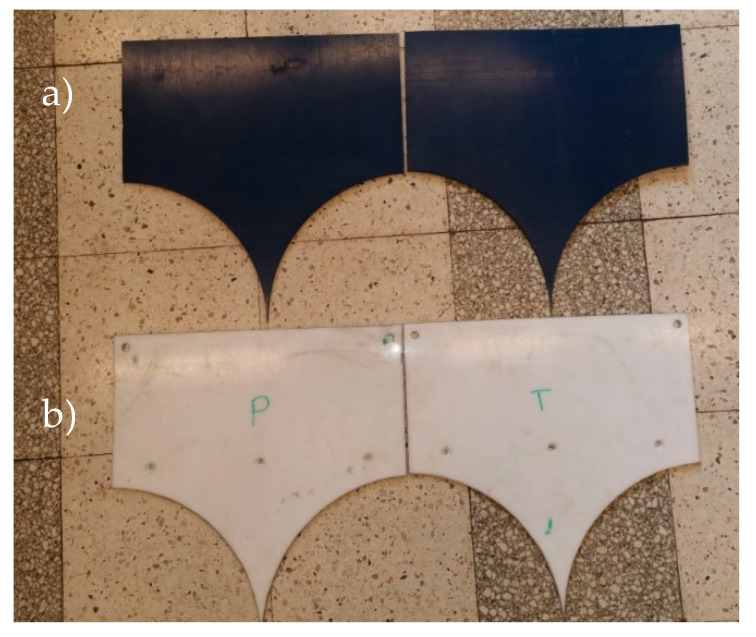
A view of cut-out slide boards: (**a**) made of the UHMW-PE material and (**b**) made of the POM C material.

**Figure 5 polymers-12-02489-f005:**
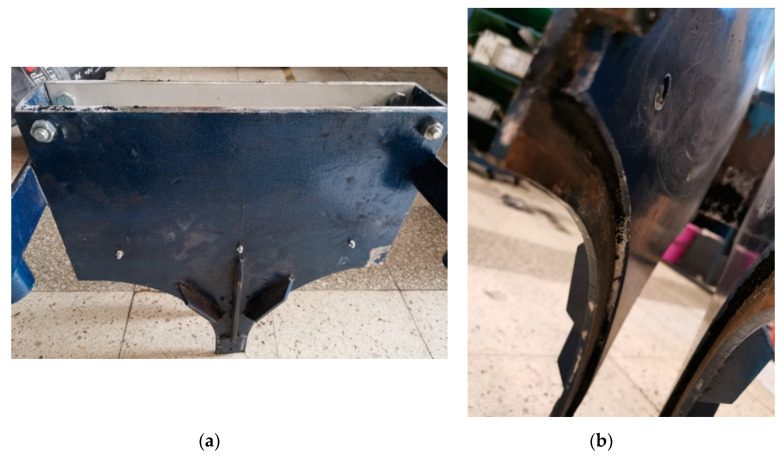
Photograph of the interior of the gravity feeder with installed boards made of the material: (**a**) POM C and (**b**) UHMW-PE.

**Figure 6 polymers-12-02489-f006:**
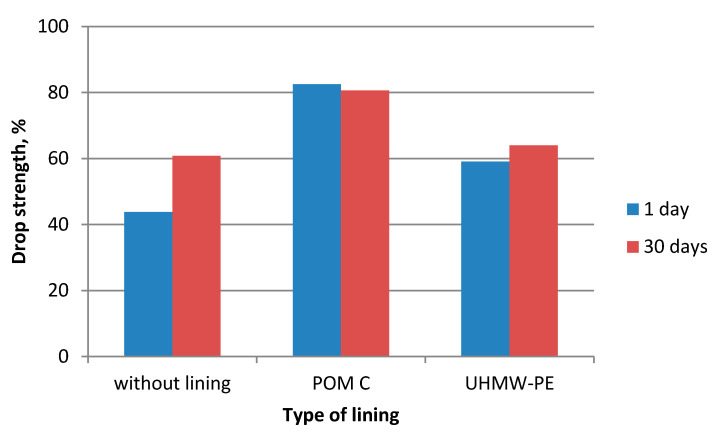
Drop strength of briquettes made of Mixture 1 (calcium hydroxide).

**Figure 7 polymers-12-02489-f007:**
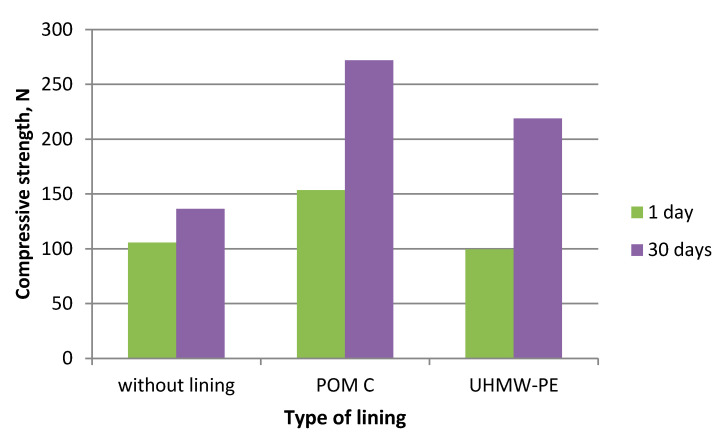
Compressive strength of briquettes made of Mixture 1 (calcium hydroxide).

**Figure 8 polymers-12-02489-f008:**
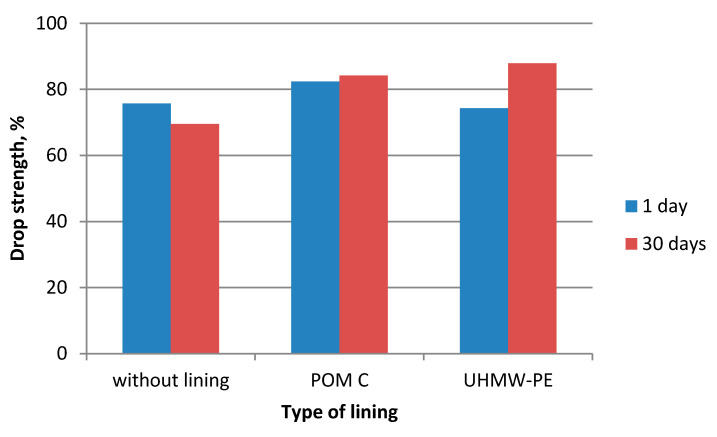
Drop strength of briquettes made of Mixture 2 (fertiliser).

**Figure 9 polymers-12-02489-f009:**
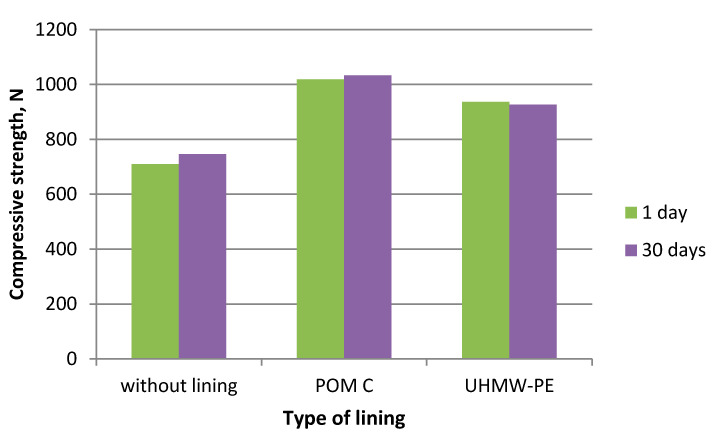
Compressive strength of briquettes made of Mixture 2 (fertiliser).

**Figure 10 polymers-12-02489-f010:**
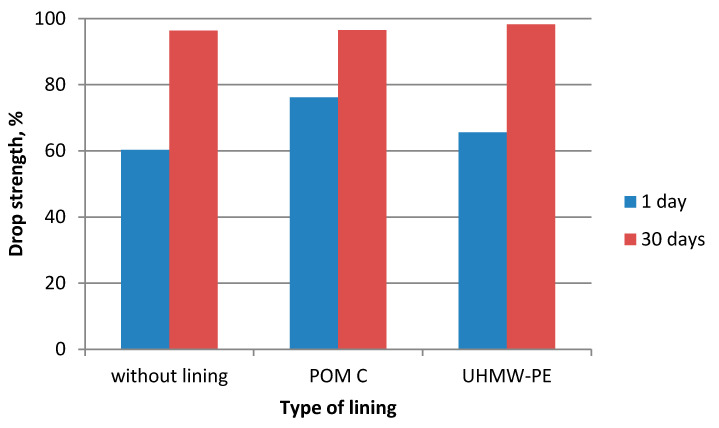
Drop strength of briquettes made of Mixture 3 (charcoal fines mixed with 4% of starch).

**Figure 11 polymers-12-02489-f011:**
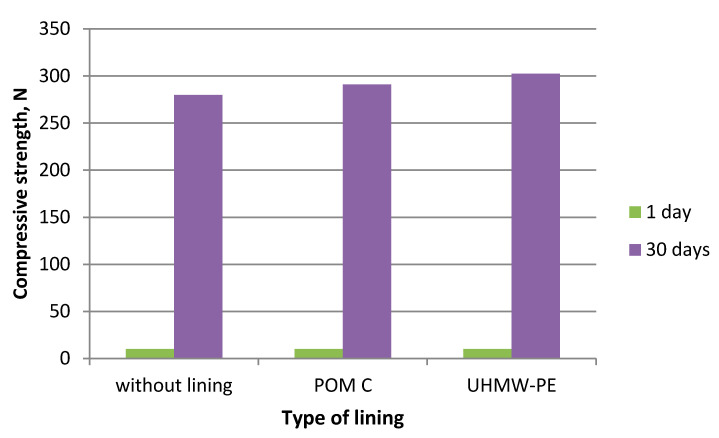
Compressive strength of briquettes made of Mixture 3 (charcoal fines mixed with 4% of starch).

**Figure 12 polymers-12-02489-f012:**
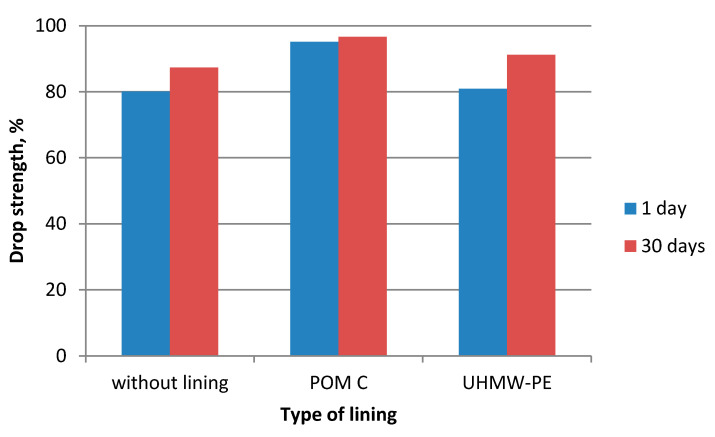
Drop strength of briquettes made of Mixture 6 (EAFD mixture).

**Figure 13 polymers-12-02489-f013:**
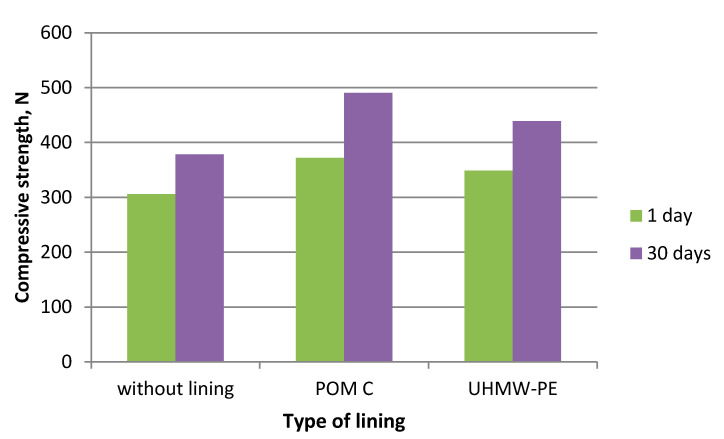
Compressive strength of briquettes made of Mixture 6 (EAFD mixture).

**Table 1 polymers-12-02489-t001:** The parameters of the POM C [49] and UHMW-PE [50] polymers used for test.

Paramert	POM C	UHMW-PE
Yield Point, MPa	67	17
Elongation at Yield point, %	9	20
Stress at Break, MPa	67	50
Impact strength with Notch (Charpy), kJ/m^2^	165	170
Melting Point, °C	166	130
Long-lasting Maximum Usable Temperature °C	140	80
Density, g/cm^3^	1.41	0.93
Water Absorption, %	0.1	0.01

**Table 2 polymers-12-02489-t002:** Results of sample surface roughness measurements.

Parameter	POM C, μm	UHMW-PE, μm
R_z_—Greatest Height of the Profile	13.88	4.91
R_t_—Total Height of the Profile	18.82	8.97
R_a_—Arithmetic Mean of the Profile Ordinates	2.56	0.73
S_m_—Average Width of Grooves of the Profile Elements	364.0	225.1

**Table 3 polymers-12-02489-t003:** Results of sample surface roughness measurements.

Parameter	Without Lining	POM C	UHMW-PE
Material 1—Calcium Hydroxide			
Drop Strength after 1 Day, %	43.8	82.5	59.1
Drop Strength after 30 Days, %	60.8	80.6	64.0
Compressive Strength after 1 Day, n	105.7	153.6	99.5
Compressive Strength after 30 Days, n	136.5	271.9	218.8
Material 2—Fertiliser			
Drop Strength after 1 Day, %	75.7	82.4	74.3
Drop Strength after 30 Days, %	69.5	84.2	87.9
Compressive Strength after 1 Day, n	709.6	1018.4	936.6
Compressive Strength after 30 Days, n	746.6	1032.8	927.0
Material 3—Charcoal Fines Mixed with 4% of Starch			
Drop Strength after 1 Day, %	60.3	76.2	65.6
Drop Strength after 30 Day, %	96.4	96.5	98.3
Compressive Strength after 1 Days, n	-- ^1^	-- ^1^	-- ^1^
Compressive Strength after 30 Days, n	279.8	290.9	302.3
Material 6—EAFD Mixture			
Drop Strength after 1 Day, %	80.1	95.1	80.9
Drop Strength after 30 Days, %	87.3	96.6	91.2
Compressive Strength after 1 Day, n	306	398.8	383.1
Compressive Strength after 30 Days, n	378.3	490.3	438.8

^1^ briquettes were of plastic nature; it was impossible to determine the compressive strength.

**Table 4 polymers-12-02489-t004:** Percent change of the compressive strength and of the average drop strength against the results obtained with no linings used.

Parameter	POM C		UHMW-PE	
	1 day	30 Days	1 Day	30 Days
Material 1—Calcium Hydroxide				
Drop Strength, %	88.4	32.6	34.9	5.3
Compressive Strength, %	45.3	99.2	−5.9	60.3
Material 2—Fertiliser				
Drop Strength, %	8.9	21.2	−1.8	26.5
Compressive Strength, %	43.5	38.3	32.0	24.2
Material 3—Charcoal Fines Mixed with 4% of Starch				
Drop Strength, %	26.4	0.1	8.8	2.0
Compressive Strength, %	-- ^1^	4.0	-- ^1^	8.0
Material 6—EAFD Mixture				
Drop Strength, %	18.7	10.7	1.0	4.5
Compressive Strength, %	21.6	29.6	13.9	16.0

^1^ briquettes were of plastic nature; it was impossible to determine the compressive strength.

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
