# Peer review of "The Influence of the Use of Polymer Lining within the Roller Press Gravity Feeder on Briquette Quality"

_polymers, 2020, doi:10.3390/polym12112489_

Round 1
Reviewer 1 Report
In this article, the author presented laboratory tests of the impact of using different polymer lining in the gravity feeder of a roller press. Two polymer were tested, POM C and UHMW-PE. The results indicated a significant improvement of polymer lining especially with POM C, compared to no treatment and UHMW-PE. The influence of the use of plates on the material flow and quality of briquettes was investigated. I recommend this article to be published in Polymers after some minor concerns are addressed:
- With mixture 3 (charcoal and starch), seems like the polymer coating with either UHMW-PE or POM C did not show much improvement to the material nonetheless. What is the reason of that?
- From the results, POM C is dominating in most of the comparisons, is there any drawbacks for POM C?
- The demical points are inconsistent among the whole article. Some use "," and some use ".". Please revise and be consistent.
Author Response
1. With mixture 3 (charcoal and starch), seems like the polymer coating with either UHMW-PE or POM C did not show much improvement to the material nonetheless. What is the reason of that?
The smallest impact of the use of polymer linings on the briquettes quality was observed for Mixture 3 (charcoal fines mixed with 4% of starch). However, it should be noted that the mixture prepared for briquetting was heated up to 70°C. This likely caused the contained heated binder in the form of starch dissolved in water to adhere more closely to the liners of the feeder.
It was added to the Discussion.
2. From the results, POM C is dominating in most of the comparisons, is there any drawbacks for POM C?
At present, the disadvantages of using this material as a lining for the feeders are not noticed.
It was added to the Discussion.
3. The demical points are inconsistent among the whole article. Some use "," and some use ".". Please revise and be consistent.
It was corrected in the article.
Reviewer 2 Report
This paper studied the impact of using a polymer plate lining in the gravity feeder of a roller press. The influence of the use of plates on the material flow and quality of briquettes was investigated. However, the quality of the paper is below the journal publication standard. This paper should be rejected due to the following reasons:
- Please give the full term of the abbreviations for the first appearance in the paper such as 'POM C' and 'UHMW-PE' in Abstract, etc.
- In Introduction, please add some detailed results from the literature.
- English writing is poor which needs major improvements such as 'results from' in Line 32, 'materials' in Line 146, etc.
- Different polymer materials should be compared and discussed in detail in Introduction.
- In Table 1, what are the materials properties which were in your setup? These parameters have been cited from other works.
- What is '0.05/0.1' in Table 1 for the water absorption percentage?
- This work mainly introduced the different materials for the roller. It is more suited to the material design and testing journals. You can check the references cited in this work. The authors only compared the results with 1 and 30 days. More data should be studied and discussed.
- There is no Conclusions section in the paper.
Author Response
1. Please give the full term of the abbreviations for the first appearance in the paper such as 'POM C' and 'UHMW-PE' in Abstract, etc.
It was corrected in the article.
2. In Introduction, please add some detailed results from the literature.
There is no data about using linings in the feeders.
3. English writing is poor which needs major improvements such as 'results from' in Line 32, 'materials' in Line 146, etc.
It was corrected in the article
4. Different polymer materials should be compared and discussed in detail in Introduction.
The article focuses mainly on polymer abrasion resistant materials used for liners. These materials are used in tanks and bunkers.
5. In Table 1, what are the materials properties which were in your setup? These parameters have been cited from other works.
The material properties have been cited from Product Information Sheet and based on DIN EN ISO 527 norm.
6. What is '0.05/0.1' in Table 1 for the water absorption percentage?
It was corrected in the article
This work mainly introduced the different materials for the roller. It is more suited to the material design and testing journals. You can check the references cited in this work. The authors only compared the results with 1 and 30 days. More data should be studied and discussed.
Briquetted materials depend on their properties and the purpose of using them in the briquetted form need different seasoning time. In some cases the seasoning is not necessary. To compare the different type of materials it was necessary to uniform the seasoning time. One day and 30 days according to my knowledge seemed universal to all of this material types.
7. There is no Conclusions section in the paper.
The Conclusion has been added
Reviewer 3 Report
In the manuscript, the influence of the use of polymer lining within the roller press gravity feeder on briquette quality. The point is very novel, interesting and scientifically important to readers. The present manuscript is acceptable.
Author Response
Dear Reviewer #3,
Thank you very much for reviewing my article.
Best regards,
Michał Bembenek.
Round 2
Reviewer 2 Report
In Introduction, please avoid "lump sum references", such as XXXXX [1-5]; all references should be cited with detailed and specific description.
The novelty of your work should be highlighted compared with the previous publications.
Author Response
Dear Reviewer #2
Thank you very much for the re-review and all comments to the article.
The article was corrected according to yours suggestion.
- In Introduction, please avoid "lump sum references", such as XXXXX [1-5]; all references should be cited with detailed and specific description.
I corrected it in the article.
- The novelty of your work should be highlighted compared with the previous publications
I put the information on the end of Conclusion that the researches have the innovative character.
Best regards,
Michał Bembenek.